# Tissue vs. Fecal-Derived Bacterial Dysbiosis in Precancerous Colorectal Lesions: A Systematic Review

**DOI:** 10.3390/cancers15051602

**Published:** 2023-03-04

**Authors:** Jurate Valciukiene, Kestutis Strupas, Tomas Poskus

**Affiliations:** Clinic of Gastroenterology, Nephro-Urology, and Surgery, Institute of Clinical Medicine, Faculty of Medicine, Vilnius University, 03101 Vilnius, Lithuania

**Keywords:** intestinal microbiota, colorectal adenoma, colorectal neoplasm, polyp, gut, carcinoma in situ, tissue-derived, fecal-derived dysbiosis, mucosa samples, stool, bacteria

## Abstract

**Simple Summary:**

Although alterations of intestinal bacterial microbiota have been admitted as playing one of the most important roles in colorectal carcinogenesis, the links between microbiota compositional changes and premalignant colorectal polyps have still not been fully examined. Furthermore, there is a lack of knowledge in terms of defining the precise differences and the correct interpretation of tissue-derived and stool-based bacterial dysbiosis in patients with precancerous colorectal lesions. Thus, this systematic review aims, firstly, to assess whether and how the tissue-derived intestinal microbiota structure differs from the bacterial dysbiosis in fecal samples of patients with simple, advanced colorectal adenoma and carcinoma in situ, and, secondly, to propose the correct selection of each matrix in order to increase sampling accuracy and applicability in future microbiota studies and clinical practice.

**Abstract:**

Alterations in gut microbiota play a pivotal role in the adenoma-carcinoma sequence. However, there is still a notable lack of the correct implementation of tissue and fecal sampling in the setting of human gut microbiota examination. This study aimed to review the literature and to consolidate the current evidence on the use of mucosa and a stool-based matrix investigating human gut microbiota changes in precancerous colorectal lesions. A systematic review of papers from 2012 until November 2022 published on the PubMed and Web of Science databases was conducted. The majority of the included studies have significantly associated gut microbial dysbiosis with premalignant polyps in the colorectum. Although methodological differences hampered the precise fecal and tissue-derived dysbiosis comparison, the analysis revealed several common characteristics in stool-based and fecal-derived gut microbiota structures in patients with colorectal polyps: simple or advanced adenomas, serrated lesions, and carcinomas in situ. The mucosal samples considered were more relevant for the evaluation of microbiota’s pathophysiological involvement in CR carcinogenesis, while non-invasive stool sampling could be beneficial for early CRC detection strategies in the future. Further studies are required to identify and validate mucosa-associated and luminal colorectal microbial patterns and their role in CRC carcinogenesis, as well as in the clinical setting of human microbiota studies.

## 1. Introduction

With more than 1.9 million new cases and 935,000 deaths, colorectal cancer (CRC) is the third most diagnosed and the second leading cause of death among cancers worldwide [1]. Several risk factors are associated with the development of CRC through a conventional adenoma-carcinoma sequence and serrated pathways. Such factors include genetical mutations, environmental, lifestyle, and dietary habits. Nevertheless, compositional changes in gut microbiota and a shift in the diversity and distribution of bacterial communities determine increased mucosal permeability, bacterial translocation, and the activation of the immune system, causing chronic inflammation and CRC [2,3,4,5].

The collection of bacteria, archaea and eukarya colonizing the human GI tract is termed the human gut microbiota, while the entire habitat (intestines), including the microorganisms, their genes, and the surrounding environmental conditions, is commonly called the gut microbiome. As for gut bacterial dysbiosis, it describes the altered composition and reduced diversity of core bacterial communities living in the gut [2,4].

Recently, several clinical trials investigating the role of intestinal bacterial dysbiosis in the stages of colorectal carcinogenesis have been published [4,5,6,7]. Multiple studies have shown a strong link between alterations in the human intestinal microbiota and the presence of carcinoma lesions in the colorectum [5,6,7,8,9]. Furthermore, some of the microbes, such as *Fusobacterium nucleatum* (*F. nucleatum*), *Streptococcus gallolyticus* (*S. bovis*), *enterotoxigenic Bacteroides fragilis* (*ETBF*), *pks* (polyketide synthase) *+ Escherichia coli*, *Enterococcus faecalis*, *Peptostreptococcus anaerobius* and *Parvimonas micra*, etc., were accepted as CRC-associated bacteria [10,11,12,13,14,15]. Colorectal adenoma (CRA) is a precancerous lesion of CRC, and recent research also finds it to be associated with an altered gut bacterial community structure, a lower richness, and a higher abundance of proinflammatory bacteria [15,16,17,18,19].

While emerging evidence suggests a link between the gut microbiota and CRC, it is hard to say that certain bacteria play an exceptional causal role in CRC, where secondary alterations in the local gut microbiota due to chronic inflammation dominate. In contrast, CR adenoma as a premalignant state does not induce severe local inflammation and consequent changes in the gut microbiota. Therefore, the indisputable detection of a statistically significant correlation between adenomatous colorectal (CR) polyps, as an early stage of the adenoma-carcinoma cascade, and intestinal microbial dysbiosis, would potentially imply primary microbiota’s role in CR tumorigenesis [20].

Sample collection is another challenging step in human microbiome studies. With the expansion of research in the field, advanced invasive and non-invasive examination models have been engaged in the detection of CRC-associated bacteria and the overall bacterial composition in human samples. Since the 1990s, molecular tools targeting the bacterial 16S ribosomal RNA (rRNA) gene have been applied for the explicit evaluation of the gut microbiota from both feces and tissues. Although both tissue and fecal specimens provide useful information about the composition of gut bacterial communities, most of the studies on the gut microbiome, including those related to CRC and CR adenoma, are still based only on fecal samples, as an easy and non-invasive procedure [21,22]. On the contrary, other clinical trials find no statistically significant association between premalignant CR polypoid lesions and an increase in CRC-associated bacteria in stool samples compared with gut mucosal biopsies, where an increased *F. nucleatum* abundance has been recognized [23]. Similarly, while some researchers state that bacterial community compositions in feces and mucosa differ completely [23,24], others believe that similar variations in CRC bacterial species can be identified between stool samples and gut mucosal biopsies [25,26]. Moreover, several studies suggest that tissue samples are more relevant for the evaluation of microbiota’s pathophysiological involvement in CR carcinogenesis, while stool samples are more powerful for identifying noninvasive diagnostic or prognostic markers of CRC [23,24,27].

Thus, a discussion arises from the detection of different intestinal microbiota shifts in mucosal and fecal samples of patients with CR neoplasia. Moreover, the proper employment of a sampling matrix and its accurate analysis for the examination of intestinal microbiota’s structural and functional composition is still lacking.

Therefore, there is a need for a profound systematic literature review, firstly, to assess the difference between mucosa-associated (tissue) and luminal (fecal) intestinal microbiota alterations in patients with precancerous colorectal lesions (simple and advanced conventional adenoma, serrated adenoma) and preinvasive cancer (carcinoma in situ (Ca in situ)), compared with healthy control and/or self-control groups, and, secondly, to suggest the potentially correct implementation and assessment of tissue and stool samples in human gut microbiota studies and in the clinical field. The following could induce a new research era based on a comprehensive methodology and the accurate use of a selected type of matrix for the precise analysis of CR carcinogenesis. This would also contribute to the validation of preventive measures for the early detection of colorectal neoplasms, as well as the management of the affected gut microbiota in premalignant mucosal changes prior to the development of CRC.

## 2. Materials and Methods

The present systematic review was performed according to the Cochrane collaboration-specific protocol [28] and was reported following the Preferred Reporting Items for Systematic Reviews and Meta-analyses (PRISMA) checklist [29]. The PRISMA checklist was completed according to the mentioned recommendations (Appendix A). The present systematic review was prospectively registered in PROSPERO (ID No.: CRD42022376106).

### 2.1. Eligibility Criteria

Studies that examined the link between CRA or Ca in situ, as precancerous colorectal lesions, and the intestinal tissue- and/or fecal-derived intestinal microbiota composition were eligible for inclusion. The search was restricted to human studies published in the English language from 2012 until November 2022. These studies included adult patients (≥18 years) with a diagnosed CR advanced/non-advanced adenoma, serrated polyps, or Ca in situ undergoing a complete examination of the tissue and/or stool-based bacterial microbiota community structure. Advanced adenomas were defined as those with high-grade dysplasia, villous or tubulovillous histology, or a diameter ≥1 cm, while serrated polyps included sessile serrated (SSA) and traditional serrated adenomas (TSA). The analysis included only studies with a complete bacterial community assessment and healthy or the same patients’ paired normal samples as controls.

### 2.2. Information Sources

A literature search was performed in the PubMed and Web of Science online databases to identify original comparative studies analyzing the colorectal mucosa-associated and/or luminal microbiota composition in patients with premalignant (adenoma) and preinvasive (Ca in situ) colorectal neoplasia. The most recent search was performed in November 2022.

### 2.3. Literature Search Strategy

We used the following combination of Medical Subject Headings (MeSH) and keywords with the employment of “AND” or “OR” Boolean operators: “Colorectal adenoma” OR “Colorectal polyp” OR “Colorectal polypoid lesion” OR “Colorectal precancerous lesion” OR “Colorectal neoplasms” OR “Colorectal neoplasia” OR “Colonic neoplasia” AND “Serrated adenoma” OR “Serrated polyp” AND “Colorectal carcinogenesis” OR “Colorectal tumorigenesis” OR “Adenoma-carcinoma sequence” AND “Gut microbiome” OR “Gut microbiota” OR “Intestinal microbiome” OR “Intestinal microbiota” OR “Gut dysbiosis” OR “Intestinal dysbiosis” AND “Mucosa-adherent” OR “Tissue-derived” OR “Mucosa-associated” OR “Stool-based” OR “Luminal” OR “Fecal-derived”.

### 2.4. Study Selection

All titles and abstracts identified in the electronic databases were screened by two experienced reviewers independently of one another using a piloted electronic database (Microsoft Excel). Following the identification of relevant abstracts, full-text articles were retrieved and re-reviewed. Comments on articles, short notes, letters, conference abstracts, systematic reviews, meta-analyses, review articles, preclinical studies, and duplicates were manually excluded. A manual search was performed to identify additional primary studies and minimize search bias. The literature review was completed with an extensive search using the “related articles” function in PubMed. Studies which did not analyze tissue- and/or stool-based bacterial gut microbiota’s structure in patients with colorectal premalignant or preinvasive neoplasms in comparison with healthy or self-controls and/or CRC were excluded. The endpoint measures of the current review consisted of tissue and fecal-derived microbiota’s bacterial compositional diversity in CRA and Ca in situ.

### 2.5. Data Extraction

The following data were extracted from each study: first, the author’s name, the date of publication, the sample size, including the number of cases and controls, the microbiota examination method and the matrix type, the abundance and/or prevalence of CRC-associated and other bacteria, α- and β-diversity, and the other main findings of the study. The term α-diversity was described as the variation of microbes in a single sample and expressed by richness (that is, the number of taxa present in a sample) and evenness (that is, how evenly distributed the taxa are within a sample). Contrarily, β-diversity was determined by the variation in microbial communities between the samples in terms of ecological distance, likely reflecting the presence and absence of some rare species. The extracted data was only evaluated at the end of the reviewing process to reduce the selection bias.

### 2.6. Study Quality Assessment and Risk of Bias

The methodological quality of the selected trials was assessed using the Cochrane Handbook method [28]. For evaluating the quality of non-randomized trials, items of risk in patient selection, baseline comparability, and outcomes/exposure selection and measurement were judged using the Newcastle–Ottawa scale (NOS) [30]. We rated the quality of the studies by awarding stars in each domain as follows: a “good” quality score required 3 or 4 stars in selection, 1 or 2 stars in comparability, and 2 or 3 stars in outcomes; a “fair” quality score required 2 stars in selection, 1 or 2 stars in comparability, and 2 or 3 stars in outcomes; a “poor” quality score was reflected by 0 or 1 star(s) in selection, or 0 stars in comparability, or 0 or 1 star(s) in outcomes. Only good and fair quality studies were included in the further analysis. A summary of the quality evaluation process has been visualized in Table 2 and Appendix A.

## 3. Results

### Search Results and Study Characteristics

The initial search yielded 292 results; after removing duplicates, 286 articles were screened for eligibility based on the title and abstract, and 64 articles were retrieved for a full-text evaluation. These were assessed for eligibility. A total of 35 were excluded as ineligible for inclusion: 4—review articles, 3—editorials, 1—video vignette, 3—conference abstracts, 5—inadequate data, 3—overlapping data, 4—no appropriate control, and 12—animal/cells study. Three studies were included additionally after the search update. All the included studies were observational: cohort, cross sectional, and case-control studies. No randomized control trials were identified. A total of 32 studies fulfilled the inclusion criteria and were finally selected for a qualitative analysis (Figure 1).

The included studies were grouped according to the utilized samples and study goals, as follows: (a) studies investigating the composition of gut bacterial tissue-derived microbiota (*n* = 11), (b) studies examining the structure of gut bacterial stool-based microbiota (*n* = 14), and (c) those investigating the composition of both tissue- and/vs. fecal-derived gut bacterial microbiota (*n* = 7) in CRA and Ca in situ.

Most of the studies included used the same ‘human gut microbiota’ term for the evaluation of bacterial communities prevailing in the gut, and the term ‘bacterial dysbiosis’ for the examination of gut bacterial composition and diversity changes. Very few studies referred to the ‘microbiome’ [27,31,32,33,34] and ‘metabolome’ [32,35,36] as study outcomes, and these generally served as a data supplementing factor for the review.

Of the studies included, the comparison of the microbiota was often between conventional adenoma without further specification of type [18,23,24,25,27,30,33,36,37,38,39,40,41,42,43,44,45,46,47,48] or adenoma classified as advanced and non-advanced [32,34,35,49,50,51,52,53], and healthy controls. However, there were few studies that investigated the microbiota’s composition in sessile serrated polyps (SSPs) and traditional serrated adenomas (TSAs) in the serrated pathway specifically [31,51,54]. Only one study aimed to compare microbiota between patients with Ca in situ vs. healthy controls [55].

Eleven studies used DNA analysis with 16S rRNA gene sequencing [27,33,35,37,38,40,45,48,49,51,52] and one was a metagenome-wide association study (MGWAS) [50]. One study used shotgun metagenomic sequencing [39], one used qPCR with liquid (LC−TOFMS) and gas (GC−TOFMS) chromatography time-of-flight mass spectrometry [36], and one used terminal restriction fragment length polymorphism (T-RFLP) with next-generation sequencing (NGS) [55]. The remaining study used ENTERO-test 24 plus MALDI-TOF mass spectrometry [46]. Five other studies also used 454-pyrosequencing [18,25,41,43,44], while five used qPCR [23,24,25,26,47], one used high-throughput sequencing (HTS) and fluorescence in situ hybridization (FISH) [42], one used PCR and FISH [54], and one used only HTS [53]. One study used high-performance liquid chromatography (HPLC) [34], one used ultra-performance liquid chromatography-mass spectrometry (UPLC-MS) metabolomics [32], one used internal transcribed spacer (ITS) ribosomal RNA sequencing and whole-genome shotgun sequencing (WGS) [31], and one used metagenomic sequencing [30] in addition to 16S rRNA gene sequencing.

All the included studies investigated the association between gut bacterial dysbiosis in fecal and/or intestinal tissue samples and precancerous colorectal lesions (and/or CRC). Two parts of the included studies additionally examined the following: the relationship between metabolites and the metagenome-metabolome [32,34,35,36]; genetic mutations [48]; the presence of mucosal biofilm [42]; diagnostic biomarkers [26,27,33,39,40,52,53]; enterotypes and clusters [24,38]; the intracellular microbiota structure [46]; location-specific microbiota [24,54]; “on” and “off” tumor bacterial differences [24]; and the fungal community composition [31] in patients with colorectal polyps.

Twelve studies were evaluated as representative with an estimate of more than 100 subjects per case group.

Most of the studies used tissue specimens directly from the lesion [23,26,42,43,44,45,46,47,48,53,54], while others preferred non-tumor colon or rectal mucosa sampling [25,36,41] for the case group analysis. In several studies both “on” and “off” tumor sampling was planned [24,27,30,31]. Seventy-eight percent of the included studies used stool samples and/or intact colorectal mucosa specimens from healthy patients for the control group [18,23,24,25,26,27,30,31,32,33,34,35,36,37,38,39,40,41,43,45,49,50,51,52,54,55]. In other studies, paired adjacent normal mucosa samples were employed for self-control group formation [47,48]. The remaining studies used both controls [42,44,46,53]. One trial also included adenoma up to 1 cm in addition to neoplasia-free colon tissue sampling for the control group [30].

The characteristics and outcomes of the studies are displayed in Table 1. Additionally, the extended version of the table explicitly describing the type of matrix, the gut microbiota’s structure, and its compositional shifts in the fecal and tissue samples of patients with colorectal adenoma, and/or colorectal cancer vs. healthy controls is provided in a Appendix A.

Based on the NOS assessment [56], 16 studies had a score of 5/9, 9 studies scored 6/9, and 7 studies scored 7/9 (Table 2). Overall, a high heterogeneity was observed in the study designs, study populations, and the examination methods of the gut microbiota’s composition.

## 4. Discussion

### 4.1. Structural Gut Microbiota Profile in Patients with Precancerous and Preinvasive Colorectal Lesions vs. HC

This systematic review revealed certain differences in the gut microbiota diversity and abundance of bacteria in patients with colorectal adenomas and Ca in situ compared to healthy adults. This is supported by most of the included studies. However, several studies reported no significant difference in microbiota diversity [45,48,52], while others did not report any difference in the microbiota’s bacterial composition between subjects with precancerous colorectal lesions and healthy controls [32,55]. Among microbiota’s compositional alterations, the most common were an increased abundance of *Fusobacterium*, *Escherichia*-*Shigella*, *Coprococcus*, *Streptococcus*, *Enterococcus*, and/or *Ruminoccocus* spp. and a reduction in *Actinobacteria*, *Firmicutes*, *Eubacteria*, *Bifidobacterium*, *Lactobacillus*, and butyrate-producing bacteria (*Clostridium*, *Roseburia*, *Eubacterium*, *Blautia*, and *Dorea* spp). These were evident in both mucosal and fecal samples in colorectal adenoma vs. healthy controls [23,25,27,37,49,51,55]. No consensus in the α-diversity and β-diversity was evident between the patterns of tissue- and fecal-derived microbiota in preneoplastic colorectal lesions. Overall, a reduction in the diversity/richness of bacterial species in the intestinal microbial community was detected in both tissue and stool samples.

While comparing stool-based microbiota composition between the case and control, eight bacterial species (*Actinomyces odontolyticus*, *Bacteroides fragilis*, *Clostridium nexile*, *Fusobacterium varium*, *Heamophilus parainfluenzae*, *Prevotella stercorea*, *Streptococcus gordonii*, and *Veillonella dispar*) and four bacterial genera (*Actinomyces*, *Atopobium*, *Fusobacterium*, and *Heamophilus*) were significantly associated with the Ca in situ group [55]. Here, the control group significantly differed with the predominant genus being *Slackia* and sp. *Eubacterium coprostanoligens*, which is a cholesterol-reducing bacteria and potentially acts as an inhibitor of CRC. However, this observational study was the only one included in the review that examined microbiota changes in patients with colorectal Ca in situ. In addition, there were only six patients forming the case group, leading to some debate on its scientific weight.

Laterally spreading tumors (LSTs) as primary precursors of CRC, due to its special morphology and growth pattern, are extremely difficult to identify during a simple colonoscopy. Thus, there is a need for new sensitive early detection methods, e.g., fecal biomarkers. Interestingly, LSTs are rarely investigated in the light of microbiota signatures. For instance, Shen et al. have demonstrated an increased fecal abundance of the three bacteria *ETBF*, *Peptostreptococcus stomatis (P. stomatis)*, and *Parvimonas micra (P. micra)* with considerably high sensitivity and specificity in detecting LST, while tissue-derived microbiota’s composition was shown to be associated with an increase in genus *Lactobacillus-Streptococcus* and the spp. *ETBF–P. stomatis–P. micra* [26]. These oral bacteria are defined as early noninvasive biomarkers of LSTs and potentially could also predict the adenoma recurrence risk after resections.

It is worth noting that the number of included studies that differentiated non-advanced adenomas (NAA) from advanced (AA) [32,34,35,49,50,51,52,53], and conventional adenomas from sessile serrated polyps (SSPs) and traditional serrated adenomas (TSAs) was rather low [31,51,54]. Different quantities of bacterial abundance were present at AA in comparison to NAA and HC. Several of the included studies demonstrated a statistically significant decrease in the butyrate producing bacteria, *Roseburia*, *Eubacteria*, and *Clostridia* [49]. Others found a considerable increase in *Fusobacterium*, *Enteroccocus*, and *Bacteroidetes* [50], while *Firmicutes* phylum and the *Firmicutes: Bacteroidetes* ratio were depleted in fecal samples of AA, though with no significant difference among the three groups (AA, CRC, and HC) [32].

Hale et al., with an estimated 780 patients included in their trial, reported a statistically significant increase in four genera: *Bilophila*, *Desulfovibrio*, *Sutterella*, and *Mogibacterium* in the stool samples of patients with diagnosed AA compared to healthy individuals. *Bilophilia* and *Desulfovibrio* are known to produce H2S and secondary bile acids, which act as a catalyzer in the A development of CRC [35]. A consistent increase in the genera *Fusobacterium*, *Tyzzerella 4*, *Phascolarctobacterium*, *Clostridium sensu stricto 1*; *Streptococcus*, *Gemella*, *Actinomyces*, and *Terrisporobacter* was observed in the fecal microbiota signatures of AA patients vs. healthy controls in a large sample size (n = 758) study. These microbial patterns could potentially supplement fecal immunohistochemical tests for the early non-invasive detection of CRA [52].

The most prominent change in colon tissue specimens of AA vs. healthy controls revealed increased *Halomonadaceae* and *Shewanella algae* and depleted *Coprococcus* and *Bacteroides ovatus* [53]. Peters et al. divided lesions into proximal and distal, AA and NAA, conventional adenomas (CA) and serrated polyps (SP). Their results showed a lower richness in CA, and especially in AA, and an enrichment of the genera *Actinomyces* and *Streptococcus* and a decrease of *Erysipelotrichi* and *Clostridia* in SSA compared to healthy controls. Colorectal serrated lesions were linked to the proximal colon location and microbiota dysbiosis was directly dependent on the severity of the lesion along the adenoma-carcinoma sequence and serrated pathway [51]. An increase in the abundance of the genus *Fusobacterium* was observed in people with serrated colorectal lesions in the reviewed studies [51,54], which was consistent with the literature on *F. nucleatum*’s primary role in the serrated neoplasia pathway [57].

Moreover, the analysis revealed well-known variations in the CRC-related bacteria found in tissue and stool samples. *Lactobacillales* were enriched in tumor tissue, *Fusobacterium*, *Porphyromonas*, *Peptostreptococcus*, *Gemella*, *Mogibacterium*, and *Klebsiella* were present in mucosa-adherent flora, while *Erysipelotrichaceae*, *Prevotellaceae*, and *Coriobacteriaceae* were highly abundant in the gut lumen of CRC patients. These prevailing bacterial communities may be related to secondary alterations in the microenvironment of CRC rather than playing a primary role in colorectal carcinogenesis. In contrast, precancerous colorectal lesions, having fewer genetic mutations and only subtle biochemical mucosal changes, have more potential to relate to the discovery of dysbiosis with initiation and acceleration processes in CRC development. Either way, these microbial signatures may resemble those presumably less severe microbiota compositional changes in precancerous colorectal lesions, adjacent tissue, and colon lumen [49,58].

The recent meta-analysis from Mo et al. concluded that the dysbiosis of the off-site (adjacent) tissue in CRC is distinctive and predictive. Tumor-adjacent tissue should not be regarded as healthy tissue and should not be used for self-controls, especially without a healthy control group [59]. In our systematic review, only a few studies employed paired adjacent tissue as self-control samples [47,48], while others used self-controls in addition to healthy patient control groups [42,44,46,53]. Several trials used normal rectal or colon mucosa samples for the case group formation [25,36,41].

Another controversial issue is the formation of biofilm in the colorectum. Biofilm is known as aggregations of microbial communities in a polymeric matrix that adhere to either biological or nonbiological surfaces, especially the colonic mucus layer coming into close contact with the mucosal epithelium itself. This contact eventually leads to altered epithelial functions and procarcinogenic tissue inflammation. One study included in the systematic review revealed a clear association between the presence of biofilm and diminished colonic epithelial cell E-cadherin, enhanced epithelial cell IL-6, and Stat3 activation. Moreover, biofilms were detected not only in tumors, both CRA and CRC, but also on tumor-free mucosa far distant from the tumors. Biofilm detection correlated with bacterial tissue invasion and changes in tissue biology with activated cellular proliferation and microbial dysbiosis [42].

### 4.2. Gut Microbiota Compositional Patterns in Mucosa-Associated (Tissue) vs. Luminal (Fecal) Samples of Patients with Premalignant Colorectal Polyps

Though the findings were inconsistent, ultimately, the majority of the studies reported statistically significant changes in microbial communities in patients with preneoplastic lesions after examining both tissue and stool samples. Regarding gut microbiota patterns and diversity in fecal vs. tissue samples in people with premalignant colorectal lesions compared to healthy controls, the results remain inconclusive. Several of the included studies declared similar variations in the microbial communities [25,26], while others reported fecal and mucosa-associated microbiotas to be completely distinct and different in composition and diversity [23,24,27]. The most common microbial signatures are displayed in Figure 2.

### 4.3. Intestinal Microbiota Studies in Patients with Precancerous CR Lesions: Tissue or Stool?

The current literature links microbial dysbiosis with CRC and colorectal precursor lesions. Through exploring the gut microbiota’s structural composition, interactions with the genome, immunome and metabolome, the main goal is to enable the creation of novel and tailored prevention, screening, and therapeutic interventions [59]. According to the included studies’ aims, objectives and outcomes, there is a certain methodology and distinct recommendations for the right selection of matrix. Here we list the main pros and cons for each type of the aforementioned specimens used in gut microbiota studies (Figure 3). 

### 4.4. Limitations of the Review

All the studies included were observational. We did not identify any randomized controlled trials which would meet the eligibility criteria and would be positively quality evaluated for inclusion. The efforts to avoid bias could have been hindered by the fact that non-English trials were not included in the review. Moreover, due to the low number of studies examining gut microbiota composition in both, tissue and/or/vs. fecal samples, we additionally included trials investigating mucosa-associated microbiota alone and those with the aim of examining luminal microbiota (marked accordingly, see Table 1). Similarly, considering the small number of studies looking only at precancerous lesions, we included those looking at premalignant lesions, preinvasive cancer, and CRC along the adenoma-carcinoma sequence. In addition, the results could be hampered by the different study sample sizes, different study populations (according to age, gender, diet, BMI, geographic location, and behavioral factors such as smoking, alcohol consumption, physical activity), different controls (healthy and/or paired normal tissue as self-controls), and different methodologies for the examination of the microbiota composition. The outcomes between the trials, including microbial diversity as well as the abundance of bacteria at phyla, family, and genus taxonomic levels in patients with precancerous colorectal lesions, were inconsistent and at some points, incomparable. Therefore, large sample-size studies examining the composition of gut microbiota in tissue and/or/vs. fecal samples and sharing their metadata are necessary in the future.

### 4.5. Recommendations and Future Prospects

Considering the large amount of upcoming clinical trials in the field, it is time to rethink our methods and the standardization of specific research practices. There is a significant need for overall recommendations for metagenomic studies which could ensure conceptual results, better comparability, the re-use of metadata, and thus more valuable research input.

The main suggestions are as follows: seek larger sample sizes; use both stool and tissue samples; examine all stages of CRC carcinogenesis; think of both conventional and serrated pathways; continue studies in comprehensive methodology; keep the important data on the type of lesions and the site of sampling performed; consider examination which provides researchers with metabolic data (shotgun sequencing) as well; keep metadata open for the availability of the research community [19,59,60].

Complete network studies investigating the interactions among gut microbiota, diet, the metabolome, genetical alterations, and local immunity responses are paramount for better CRC diagnosis and prevention strategies [60,61,62]. Likewise, understanding the tissue- and fecal-pattern of gut microbiota structure may contribute to novel strategies, such as the early noninvasive stool-based detection of colorectal adenomas and appropriate additional treatment with pre/probiotics, or immunotherapy in people with colorectal neoplasms [63,64].

## 5. Conclusions

Emerging evidence suggests that gut dysbiosis is one of the major players in the initiation and development of CRC. Ultimately, the findings of this systematic review demonstrate that precancerous colorectal lesions are associated with alterations in gut microbiota composition in both mucosal and fecal samples, in comparison to healthy and self-controls. The majority of studies examining the tissue-associated and/vs. fecal-based structure of microbiota declare a higher presence of *Fusobacterium*, *Escherichia*-*Shigella*, *Coprococcus*, *Streptococcus*, *Enterococcus*, and/or *Ruminoccocus* spp. and a lower abundance of *Actinobacteria*, *Firmicutes*, *Eubacteria*, *Bifidobacterium*, *Lactobacillus*, and butyrate-producing bacteria (*Clostridium*, *Roseburia*, *Eubacterium*, *Blautia*, and *Dorea* spp.) in both fecal and tissue specimens, and *Faecalibacterium* in stool samples from patients with precancerous colorectal lesions compared to healthy controls. Mucosa samples are becoming more relevant in the evaluation of microbiota’s pathophysiological involvement in CR carcinogenesis, while stool samples are more powerful for identifying noninvasive diagnostic or prognostic markers in CRC. Due to the high heterogeneity in terms of methodology and sample size among the included studies, the results are inconclusive. Therefore, further studies with a larger sample size, comprehensive study design, and precise sampling selection are paramount to identify and validate tissue- and fecal-derived colorectal microbial patterns and their role in CRC carcinogenesis. Understanding both the mucosa-associated and luminal pattern of gut microbiota composition could also contribute to CRC diagnosis, prevention, and treatment.

## Figures and Tables

**Figure 1 cancers-15-01602-f001:**
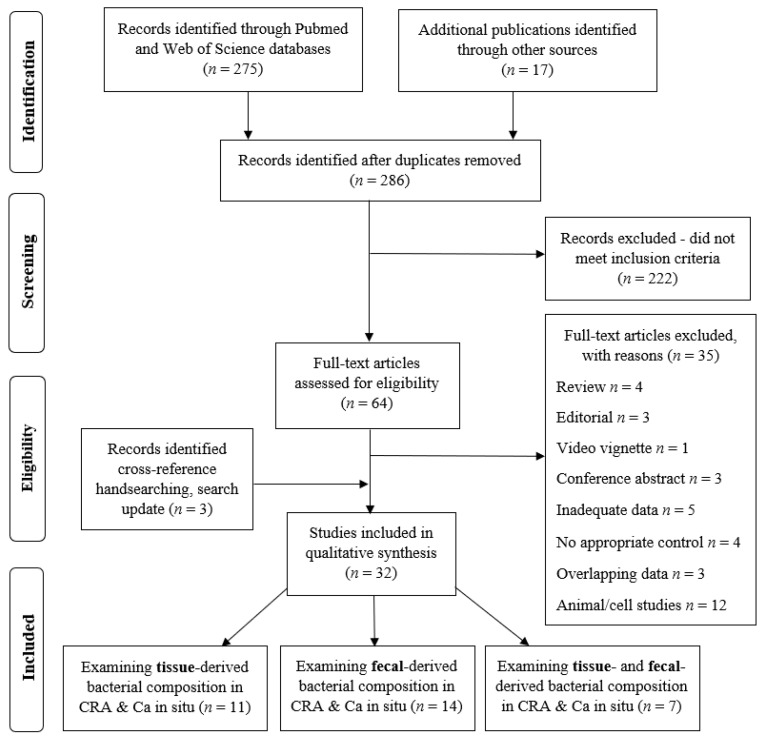
PRISMA flow diagram indicating the selection of studies for the systematic review.

**Figure 2 cancers-15-01602-f002:**
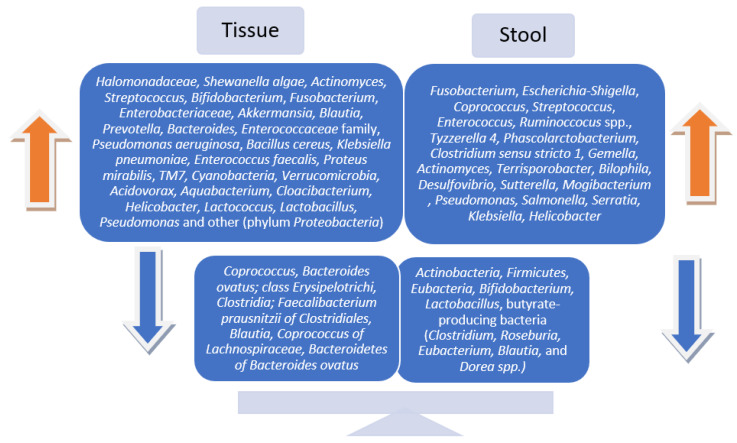
Bacterial abundances in tissue vs. stool samples in patients with precancerous colorectal lesions.

**Figure 3 cancers-15-01602-f003:**
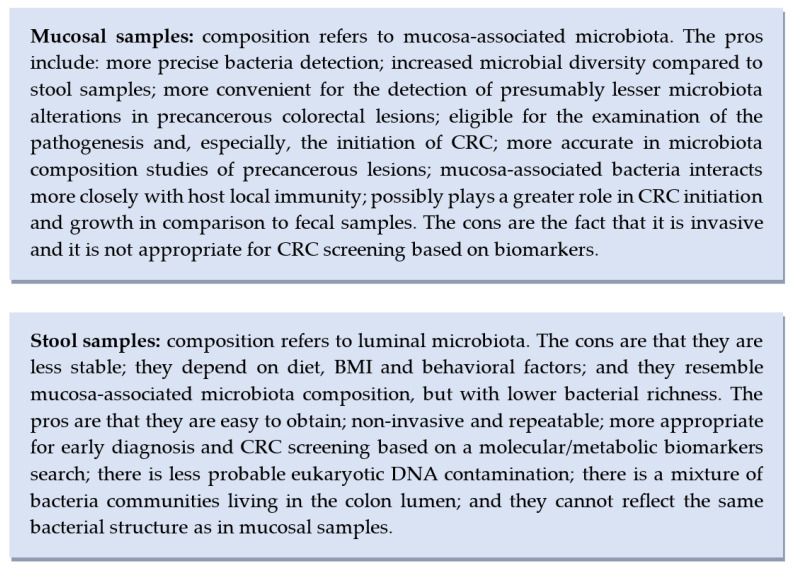
Advantages and disadvantages of mucosal vs. stool samples usage in gut microbiota studies.

**Table 1 cancers-15-01602-t001:** Summary of human studies investigating precancerous colorectal lesions and healthy control stool and tissue specimens addressing microbial compositional shifts.

Author (Publish Date)	Quality Assessment (NOS) ≥5/9	Study Group Size (n)	Control Group Size (n)	Type of Matrix (F/T)	Detection Method	Clinical Evidence
Human studies examining **FECAL** and/vs. **TISSUE**-derived gut bacterial composition in precancerous colorectal lesions (and/or CRC)						
Zeller et al. (2014) [30]	5/9	French cohort (Fr): TA: 42, CRC: 53; German cohort (G): CRC: 38, CRC: 48 (at the time of surgery)	Fr: HC: 61, A <1 cm: 27; German, Danish, and Spanish cohort (H): HC: 297	Fr and G: F; H: F; G: T	16S, metagenomic sequencing	Microbiota changes during the early stages of neoplastic growth.
Mira-Pascual et al. (2015) [23]	7/9	TA: 11; CRC: 7	HC: 10	F; T	16S: V1–V3 PCoA; *Fn* qPCR	Microbial changes according to disease progression stage and tumor severity. T samples represented the underlying dysbiosis. F samples seem not to be appropriate to detect shifts in microbial composition.
Yu et al. (2015) [25]	5/9	F: A: 47, CRC: 42; T: A: 30, CRC: 31	F: HC: 52; T: HC: 37	F; T	16S; 454 FLX pyrosequencing; *Fn* qPCR	Microbial structures were altered in the lumen and the mucosa during the progression of the A-carcinoma sequence. *Fn* expression in the T samples was consistent with that in the F samples.
Flemer et al. (2017) [24]	6/9	A: 21; CRC: 59	HC: 56 (32 age-matched)	F; T	16S; qRT-PCR	Microbiota compositional differences in patients with CRC are not secondary to the cancer per se. F microbiota only partially reflected T microbiota. T microbiota in A was similar to CRC.
Shen et al. (2021) [26]	7/9	T: A: 8, LST: 11; F: A: 208, LST: 109, CRC: 45	T: HC: 5; F: HC: 113	F; T	16S; qPCR	F microbial biomarkers *ETBF–P. stomatis–P. micra* were defined as early noninvasive biomarkers of LST.
Watson et al. (2021) [27]	5/9	A: 48	Non-A patients: 56	F; T; oral swab	16S: V4	F- and T-associated microbiomes are distinct; T microbiome is highly predictive of A status.
Avelar-Barragan et al. (2022) [31]	5/9	TA: 45; SP (HP, TSA, or SSP): 33	HC: 50	F; T	16S; ITS sequencing; WGS	Microbiomes of F samples were significantly diverse and compositionally distinct vs. mucosal aspirates. Mucosal samples are sensitive enough to study the microbiome of CRA found within the proximal colon.
Human studies examining **FECAL**-derived gut bacterial composition in precancerous colorectal lesions (and/or CRC)						
Brim et al. (2013) [18]	5/9	A: 6	HC: 6	F	16S; Human Intestinal Tract Chip; 454 pyrosequencing	*Bacteroides* group needs to be further analyzed for potential actors in the early colon oncogenic transformation.
Chen et al. (2013) [49]	5/9	AA: 47 (sex- and age-matched)	HC: 47	F	16S	A high-fiber dietary pattern, the subsequent consistent production of SCFAs, and healthy gut microbiota are associated with a decreased risk of AA.
Feng et al. (2015) [50]	7/9	AA: 44, CRC: 46 (sex-, age-, race-matched)	HC: 57	F	MGWAS	Development of AA and CRC.
Goedert et al. (2015) [37]	5/9	A: 20; CRC: 2; other: 15	HC: 24	F	16S	If confirmed in larger, more diverse populations, F microbiota analysis might be employed to improve screening for CRA.
Kasai et al. (2016) [55]	5/9	A: 50; CRC: 9 (3—invasive; 6—Cis)	HC: 49	F	T-RFLP; NGS	Gut microbiota is related to CRC prevention and development.
Peters et al. (2016) [51]	7/9	CA: 144 (proximal: 87, distal: 55, NAA: 121, AA: 22); SA: 73 (HP: 40, SSA: 33)	HC: 323	F	16S	Gut microbes may play a role in the early stages of CR carcinogenesis through the development of CAs.
Hale et al. (2017) [35]	5/9	A (>1 cm): 233	HC: 547	F	16S	*Bilophilia* and *Desulfovibrio* may produce genotoxic or inflammatory metabolites (H2S and secondary bile acids) playing a role in catalyzing A development and eventually CRC.
Yang et al. (2019) [38]	6/9	A: 117; CRC: 62	HC: 104	F	16S: V3-4	F microbiota differs along the A-carcinoma sequence and across enterotypes.
Clos-Garcia et al. (2020) [32]	7/9	AA: 69; CRC: 99	HC: 77	F	16S: V1–V2, targeted UPLC-MS metabolomics	Integration of metabolomics and microbiome data revealed tight interactions between the bacteria and host and performed better than the FOB test for CRC diagnosis.
Wei et al. (2020) [33]	5/9	A: 43	HC: 53	F	16S: V3-4, short- and long-read sequencing	Identification of adenomatous polyp-associated microbiomes could potentially function as an auxiliary biomarker for predicting CRC development.
Zhang, He et al. (2022) [39]	5/9	A: 29; CRC: 30	HC: 35	F	Shotgun metagenomic sequencing	*Peptostreptococcus stomatis*, *Clostridium symbiosum*, *Hungatella hathewayi*, *Parvimonas micra*, and *Gemella Morbillorum* were identified as a diagnostic model to identify CRC patients.
Hua et al. (2022) [40]	5/9	A: 20; CRC: 154	HC: 199	F	16S	Several intestinal bacteria changed along the A-carcinoma sequence and might be potential markers for the diagnosis and treatment of CRA/CRC.
Bosch et al. (2022) [34]	6/9	A: 32 (19 strictly matched on age, BMI and smoking habits, AA: 9; NAA: 10)	HC: 32	F	16S: V4; HPLC	The F microbiome of post-endoscopy patients resembled those of HC patients.
Zhang, Lu et al. (2022) [52]	6/9	AA: 268; NAA: 490	HC: 788	F	16S	Identified microbial signatures could complement FITs for detecting AA.
Human studies examining **TISSUE**-derived bacterial composition in precancerous colorectal lesions (and/or CRC)						
Sanapareddy et al. (2012) [41]	5/9	A: 33	A-free controls: 38	T	16S; 454 pyrosequencing	Sequence analysis of the microbiota could be used to identify patients at risk of developing A.
Dejea et al. (2014) [42]	5/9	Right-sided: A: 6, CRC: 15; Left-sided: A: 2, CRC: 15	HC: 22; paired normal adjacent tissue	T	16S: V3–V5; high-throughput sequencing; FISH	Mucosal biofilm detection correlates with bacterial tissue invasion and may predict an increased risk for the development of sporadic CRC.
Geng et al. (2014) [43]	6/9	A:10; CRC: 8	HC: 10 (location-matched)	T	16S, 454 pyrosequencing	Bacterial driver-passenger model for CRC.
Nugent et al. (2014) [36]	6/9	A: 15	A-free controls: 15	T	qPCR; LC−TOFMS; GC−TOFMS	Metabolic bacterial products and the interplay between bacteria and metabolites is important in the development of CRA and CRC.
Lu et al. (2016) [44]	7/9	A: 31	HC: 20; paired normal adjacent tissue	T	16S pyrosequencing	CR preneoplastic lesion may be the most important factor leading to alterations in the bacterial community composition.
Yu et al. (2016) [54]	6/9	Proximal HP: 35; SSA: 33; Distal HP: 40; Proximal TA: 38; Distal TA: 41; Distal CRC: 45; Proximal CRC: 48	HC: 20	T	16S; FISH; *Fn* PCR	Invasive *Fn* is involved primarily in the development of proximal colon cancers along the serrated neoplasia pathway, having only a minor role in the traditional A-carcinoma sequence.
Xu et al. (2017) [45]	6/9	A: 47; CRC: 52	HC: 61	T	16S	*Butyricicoccus*, *E. coli*, and *Fusobacterium* can be used as potential biomarkers for HC, A, and CRC groups, respectively.
Wachsmannova et al. (2018) [46]	5/9	A: 10; CRC: 10	HC: 9; paired nonmalignant tissue	T	ENTEROtest 24 plus MALDI-TOF mass spectrometry Gentamicin-protection assay	Data supports *E. coli*’s role as a pro-oncogenic pathogen.
Bundgaard-Nielsen et al. (2019) [47]	7/9	A: 96; CRC: 99; diverticular disease: 104	Paired normal tissue; No HC	T	16S; *S. gallolyticus, Fn*, *ETBF* qPCR	Findings do not support a role for *Fn* or *ETBF* during the first stages of CR, while *S. gallolyticus* was not implicated in the CR tissue of a Danish population. Potential role of the bacterial genera *Prevotella* and *Acinetobacter* requires further investigations.
Wang et al. (2020) [53]	5/9	AA: 49	HC: 36; normal adjacent tissue	T	16S: V4; high-throughput sequencing	Increasing *Halomonadaceae* and *Shewanella algae*, and decreasing *Coprococcus* and *Bacteroides ovatus* could serve as a biomarker of CRA.
Liu et al. (2021) [48]	5/9	Cohort 1: A: 10, CRC: 11; Cohort 2: A: 10, CRC: 10; +A: 12, CRC: 15	Paired normal adjacent tissue; No HC	T	16S: V4	Intra-neoplasia microbiota is heterogeneous and correlates with CR carcinogenesis.

NOS: Newcastle–Ottawa scale; NA: not applicable; A: adenoma; AA: advanced adenoma (>1 cm in diameter and/or high grade dysplasia (with or without villous, or tubulovillous morphology)); NAA: non-advanced adenoma; TA or CA: conventional or traditional adenoma (≤1 cm diameter, without dysplasia (tubular, tubulovillous, or villous)); HP: hyperplastic polyp; SA: serrated adenoma (SSA: sessile serrated adenoma or TSA: traditional serrated adenoma); Cis: Ca in situ/intramucosal carcinoma/carcinoma in adenoma; CR: colorectal; CRA: colorectal adenoma; CRC: colorectal cancer; LST: laterally spreading tumor; HC: healthy controls; F: fecal samples; T: tissue (mucosal) samples; 16S: 16s rRNA gene sequencing; WGS: whole-genome shotgun sequencing; T-RFLP: terminal restriction fragment length polymorphism; NGS: next-generation sequencing; MGWAS: metagenome-wide association study; FISH: fluorescence in situ hybridization; LC−TOFMS: liquid chromatography time-of-flight mass spectrometry; GC−TOFMS: gas chromatography time-of-flight mass spectrometry; HPLC: high-performance liquid chromatography analysis; UPLC–MS: ultra performance liquid chromatography-mass spectrometry; ITS: Internal Transcribed Spacer (ITS) ribosomal RNA sequencing; *ETBF*: enterotoxigenic *Bacteroides fragilis*; SCFAs: short-chain fatty acids; *Fn: Fusobacterium nucleatum*.

**Table 2 cancers-15-01602-t002:** Quality assessment of the selected studies according to the star score of the Newcastle–Ottawa scale (NOS), based on which * are assigned to three criteria, i.e., selection (with a maximum of 4 stars [****]), comparability between case and controls (with a maximum of 2 stars [**]), and ascertainment of effects of microbiota—outcome/exposure (with a maximum of 3 stars [***]) for a potential score ranging from 0 to 9 points. Higher scores indicate a lower risk of bias.

Author	Selection	Comparability	Outcome/ Exposure	Total Score
Tissue + Stool				
Zeller et al. (2014) [30]	**	*	**	5
Mira-Pascual et al. (2015) [23]	***	**	**	7
Yu et al. (2015) [25]	****	-	**	6
Flemer et al. (2017) [24]	***	**	*	6
Shen et al. (2021) [26]	****	-	***	7
Watson et al. (2021) [27]	***	-	**	5
Avelar-Barragan (2022) [31]	**	-	***	5
Stool				
Brim et al. (2013) [18]	***	-	**	5
Chen et al. (2013) [49]	***	**	-	5
Feng et al. (2015) [50]	***	**	**	7
Goedert et al. (2015) [37]	**	-	***	5
Kasai et al. (2016) [55]	**	-	***	5
Peters et al. (2016) [51]	****	-	***	7
Hale et al. (2017) [35]	***	-	**	5
Yang et al. (2019) [38]	****	-	**	6
Clos-Garcia et al. (2020) [32]	****	-	***	7
Wei et al. (2020) [33]	**	-	***	5
Zhang, He et al. (2022) [39]	**	-	***	5
Hua et al. (2022) [40]	***	-	**	5
Bosch et al. (2022) [34]	**	**	**	6
Zhang, Lu et al. (2022) [52]	***	-	***	6
Tissue				
Sanapareddy et al. (2012) [41]	***	-	**	5
Dejea et al. (2014) [42]	**	*	**	5
Geng et al. (2014) [43]	***	*	**	6
Nugent et al. (2014) [36]	***	*	**	6
Lu et al. (2016) [44]	***	**	**	7
Yu et al. (2016) [54]	***	*	**	6
Xu et al. (2017) [45]	****	-	**	6
Wachsmannova et al. (2018) [46]	***	-	**	5
Bundgaard-Nielsen et al. (2019) [47]	****	-	***	7
Wang et al. (2020) [53]	***	-	**	5
Liu et al. (2021) [48]	***	-	**	5

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
