# Peer review of "Tissue vs. Fecal-Derived Bacterial Dysbiosis in Precancerous Colorectal Lesions: A Systematic Review"

_cancers, 2023, doi:10.3390/cancers15051602_

Round 1

Reviewer 1 Report

The core concept of this paper is "bacterial dysbiosis". The authors need to define this term in INTRODUCTION, and tell readers whether or not all cited articles use the same/similar definition.

TABLE1 is a huge table containing a lot of abbreviations. Suggestion: put those explanations of abbreviations under each page.

Literature[37] is the only cited study in this manuscript investigating both tissue and fecal derived bacterial dysbiosis, which should earn it a special spot in DISCUSSION. 

Author Response

Point 1: The core concept of this paper is "bacterial dysbiosis". The authors need to define this term in INTRODUCTION, and tell readers whether or not all cited articles use the same/similar definition.

Response 1: ‘Human gut microbiota’, ‘gut microbiome’ and ‘bacterial dysbiosis’ terms are being defined for better clarity in INTRODUCTION (see lines 47-51).

Readers are also acknowledged about the fact that most of the included trials used the same – ‘human gut microbiota’ term for the evaluation of bacterial communities prevailing in the gut, and ‘bacterial dysbiosis’ term for alterations in gut bacterial composition. Very few studies referred to ‘microbiome’ [27, 32, 41, 42, 45] and ‘metabolome’ [39, 41, 50] as study outcomes, which was not a contradiction, but rather served as a data supplementing factor (see lines 193-197).

Point 2: TABLE1 is a huge table containing a lot of abbreviations. Suggestion: put those explanations of abbreviations under each page.

Response 2: Table 1 was replaced with simpler, shorter table presenting main characteristics of the included studies. As shortened, the abbreviations are being placed under the table only once. Additionally, the extended version of the table including data on precise microbiota’s structure and its compositional shifts is presented in supplementary material file (Table S3). Here, for better convenience, the abbreviations are put under each page, as suggested.

Point 3: Literature [37] is the only cited study in this manuscript investigating both tissue and fecal derived bacterial dysbiosis, which should earn it a special spot in DISCUSSION.

Response 3: The included study [37] is not investigating tissue and fecal derived bacterial dysbiosis. In contrary, there was only stool samples used for the examination of microbiota’s community structure and its compositional shifts. If you referred to the mentioned study, as the only one, which included carcinoma in situ (carcinoma in adenoma – as preinvasive cancer) patients, this special spot has been already analyzed in the discussion section (see lines: 272-282). Concerning studies, which included both tissue and fecal samples – all in all, there were seven studies as displayed in Figure 1, Table 1 and described in 4.2. part of the DISCUSSION, as well.

Reviewer 2 Report

The manuscript entitled "Tissue vs. Fecal-Derived Bacterial Dysbiosis in Precancerous Colorectal Lesions: A Systematic Review" presents an important topic in the human medical field. The authors produced a very well-structured, balanced and accurate presentation, the review methodology is accurate and well-described. The novelty, aim, limitations, and recommendations are well-indicated, the conclusions are pertinent.  

Only one recommendation for the authors - to check the English language to correct the grammar errors

Author Response

Point 1: Only one recommendation for the authors - to check the English language to correct the grammar errors.

Response 1: English language style and grammar were corrected where relevant (Lines 65, 78, 101, 144-145, 380, 411-413, 415, 418, 430, 433, 152, 447-448).

Reviewer 3 Report

By the review, the authors want to conclude the correlation between tissue-derived intestinal microbiota structure and the bacterial dysbiosis in fecal samples of patients with simple, advanced colorectal adenoma and carcinoma in situ, and provide the references for the future studies and clinical practice.

The reviewer think that this is a valuable work because colorectal carcinoma is a popular malignancy and its oncogenesis or progression maybe probably correlated with intestinal microbiota structure and the bacterial dysbiosis.

Major Comments

1.        Figure 1 is not so qualified for publishing.

2.        Table 1 is too long and tedious (12 pages totally)! Actually, no any reader will pay attention to this table carefully! So it is really a waste to space of pages! I strongly suggest that the authors must replace Table 1 by another simple, short and valuable manner!

3.        Figure 2 is also not so qualified for publishing.

Minor Comments

4.        It is better to figure out of the final conclusions by an illustration in “4. Discussion” section.

5.        Some qualified and important figures from references can be inserted into the text.

Author Response

Major Comments:

Point 1 and 3: Figure 1 is not so qualified for publishing. Figure 2 is also not so qualified for publishing.

Response 1 and 3: Regarding the quality problems of Figure 1 and Figure 2 pointed out in the report, it is caused by the loss of clarity when converting word to PDF. If the manuscript could be accepted, the editors will improve the figures’ clarity in the final version, as the good quality word file versions are submitted.

Point 2: Table 1 is too long and tedious (12 pages totally)! Actually, no any reader will pay attention to this table carefully! So it is really a waste to space of pages! I strongly suggest that the authors must replace Table 1 by another simple, short and valuable manner!

Response 2: Table 1 was replaced with simpler, shorter table perfectly presenting main characteristics of the included studies and its clinical evidence. Additionally, the extended version of the table explicitly describing type of matrix, gut microbiota’s structure, and its compositional shifts in fecal and tissue samples of patients with colorectal adenoma, and/or colorectal cancer versus healthy controls is provided in supplementary material file (Table S3).

Minor Comments:

Point 1: It is better to figure out of the final conclusions by an illustration in “4. Discussion” section.

Response 1: We believe it is not suitable to add another illustration to the final conclusions, as we have already included some, both summarizing the microbiota’s changes in precancerous colorectal lesions and giving instructions for appropriate sampling and matrix. This might require the permission from the authors and publishers of the original articles and would not be original.

Point 2: Some qualified and important figures from references can be inserted into the text.

Response 2: We are assured that there is no need for another additional figures being inserted into the main text.